# Properties of Plasma Clots in Adult Patients Following Fontan Procedure: Relation to Clot Permeability and Lysis Time—Multicenter Study

**DOI:** 10.3390/jcm10245976

**Published:** 2021-12-20

**Authors:** Maciej Skubera, Aleksandra Gołąb, Dariusz Plicner, Joanna Natorska, Michał Ząbczyk, Olga Trojnarska, Anna Mazurek-Kula, Monika Smaś-Suska, Agnieszka Bartczak-Rutkowska, Piotr Podolec, Lidia Tomkiewicz-Pająk

**Affiliations:** 1Department of Cardiac and Vascular Diseases, John Paul II Hospital, 31-202 Krakow, Poland; mskubera@gmail.com (M.S.); monikasmas@gmail.com (M.S.-S.); ppodolec@interia.pl (P.P.); ltom@wp.pl (L.T.-P.); 2Institute of Cardiology, Jagiellonian University Medical College, 31-008 Krakow, Poland; j.natorska@szpitaljp2.krakow.pl (J.N.); michalzabczyk@op.pl (M.Z.); 3Faculty of Medicine and Dentistry, Pomeranian Medical University, 70-204 Szczecin, Poland; olagoab13@gmail.com; 4Department of Cardiovascular Surgery and Transplantation, John Paul II Hospital, 31-202 Krakow, Poland; 5Unit of Experimental Cardiology and Cardiac Surgery, Faculty of Medicine and Health Sciences, Andrzej Frycz Modrzewski Krakow University, 30-705 Krakow, Poland; 6Center for Research and Innovative Technology, John Paul II Hospital, 31-202 Krakow, Poland; 71st Department of Cardiology, Poznan University of Medical Sciences, 61-701 Poznan, Poland; olgatroj@wp.pl (O.T.); aga.bartczak@gmail.com (A.B.-R.); 8Department of Cardiology, Polish Mother’s Memorial Hospital, Research Institute, 93-338 Lodz, Poland; qla@op.pl

**Keywords:** clot lysis time, permeation coefficient, Fontan procedure, liver dysfunction

## Abstract

Objectives: thromboembolic complications are a major cause of morbidity and mortality following Fontan (FO) surgery. It is also well established that altered FO circulation results in systemic complications, including liver and endothelium damage. We sought to evaluate whether dysfunctions of these sources of hemostatic factors may result in changes of fibrin clot properties. Methods: a permeation coefficient (K_s_) and clot lysis time (CLT) were assessed in 66 FO patients, aged 23.0 years [IQR 19.3–27.0], and 59 controls, aged 24.0 years [IQR 19.0–29.0]. K_s_ was determined using a pressure-driven system. CLT value was measured according to assay described by Pieters et al. Endothelium and liver-derived hemostatic factors along with liver function parameters were evaluated. The median time between FO operation and investigation was 20.5 years [IQR 16.3–22.0]. Results: FO patients had lower K_s_ (*p* = 0.005) and prolonged CLT (*p* < 0.001) compared to that of controls. K_s_ correlated with CLT (r = −0.28), FVIII (r = −0.30), FIX (r = −0.38), fibrinogen (r = −0.41), ALT (r = −0.25), AST (r = −0.26), GGTP (r = −0.27) and vWF antigen (r = −0.30), (all *p* < 0.05). CLT correlated with the time between FO operation and investigation (r = 0.29) and FIX (r = 0.25), (all *p* < 0.05). After adjustment for potential cofounders, TAFI antigen and GGTP were independent predictors of reduced K_s_ (OR 1.041 per 1% increase, 95% CI 1.009–1.081, *p* = 0.011 and OR 1.025 per 1 U/L increase, 95% CI 1.005–1.053, *p* = 0.033, respectively). Protein C and LDL cholesterol predicted prolonged CLT (OR 1.078 per 1% increase, 95% CI 1.027–1.153, *p* = 0.001 and OR 6.360 per 1 μmol/L increase, 95% CI 1.492–39.894, *p* = 0.011, respectively). Whereas elevated tPA was associated with lower risk of prolonged CLT (OR 0.550 per 1 ng/mL, 95% CI 0.314–0.854, *p* = 0.004). GGTP correlated positively with time between FO surgery and investigation (r = 0.25, *p* = 0.045) and patients with abnormal elevated GGTP activity (*n* = 28, 42.4%) had decreased K_s_, compared to that of the others (5.9 × 10^−9^ cm^2^ vs. 6.8 × 10^−9^ cm^2^, *p* = 0.042). Conclusion: our study shows that cellular liver damage and endothelial injury were associated with prothrombotic clot phenotype reflected by K_s_ and CLT.

## 1. Introduction

The Fontan (FO) procedure is the end-stage palliation for patients with univentricular physiology and it considerably improved survival rates [1,2]. However, most FO adults experience cardiovascular complications over long-term follow-up, and despite various prophylactic anticoagulation algorithms, thromboembolic events are a major cause of morbidity and mortality following FO surgery [3]. Numerous hypotheses were proposed to explain the hypercoagulable state in FO patients [4,5]. It was speculated that impaired function of endothelium as well as cellular liver damage, commonly observed in adults with FO physiology, might at least partly explain these unfortunate events [6,7,8,9,10]. A crucial role in the development of the above changes may be attributed to hemodynamic alternations associated with the FO subjects’ circulation, including elevated central venous pressure and diminished cardiac output [1,10,11,12].

Fibrin clot is a common feature of all hemostatic processes. Several factors, including the geometry of the fibrin network define fibrin clot architecture, which is a key determinant of the efficiency of clot lysis [13,14]. The structure of a fibrin clot can be characterized by the fiber diameter and the size of the pores in the fibrin network. Permeability of the clot, reflected by fibrin porosity and its structural design, might be determined by the permeation coefficient (K_s_). Reduced K_s_ is a typical feature of the prothrombotic fibrin clot phenotype, which is associated with faster formation of denser fibrin mesh, relatively resistant to lysis [13,15]. Whereas clot lysis time (CLT) is the test that reflects the overall fibrinolytic activity of plasma. Increased CLT value is a manifestation of hypofibrinolysis and might predispose to both venous and arterial thrombosis [16,17].

The delicate balance between formation and lysis of the fibrin fibers may be disturbed by various factors, such as qualitative and quantitative fibrinogen disorders or inflammation that alter clot structure [13]. We already showed that adult FO patients are characterized by hepatic dysfunction as well as enhanced platelet activation and endothelial injury, enhanced thrombin formation, with impaired fibrinolysis [12,18]. To the best of our knowledge, there were no other reports evaluating plasma fibrin clot properties and their relationship with liver and endothelial damage in the FO population. Therefore, the aim of this study was to assess these potential associations in adult FO patients with the two keys measures of clot properties: clot permeability and lysability.

## 2. Materials and Methods

### 2.1. Patients

In this multicenter study (Krakow, Poznan, and Lodz in Poland), we studied 66 patients who received an FO operation without current vitamin K antagonist therapy out of a total of 91 FO adults, and 59 apparently healthy control subjects matched for age and sex. Clinical data and laboratory investigations were collected prospectively during the scheduled medical appointments, between January 2018 and December 2018. The exclusion criteria for FO patients were as follows: any acute illness, malignant neoplasms, serum creatinine level > 120 µmol/L, diabetes mellitus and alcohol abuse or pregnancy. Control subjects were eligible if they had no history of thromboembolic events (vein thrombosis, pulmonary embolism, or stroke), no signs of acute infection, recent trauma, or surgery. Pregnant women or those who delivered within the previous 12 months were also excluded (Figure 1).

The study was performed in accordance with the Declaration of Helsinki and was approved by the Research Ethics Committee 130/KBL/OIL/2018. Patients provided written informed consent. Personally identifiable information of the participants was anonymized upon extraction of the relevant data for the study, and patients were coded using numbers (1, 2, or 3, and so on).

### 2.2. Laboratory Investigations

On the same day that clinical data were recorded, fasting blood samples were collected into 0.1 volume of 3.2% trisodium citrate from the antecubital vein with minimal stasis. In anticoagulated patients, blood was drawn at least 5 days after anticoagulation withdrawal. Citrated blood samples were centrifuged at 3000× *g* for 20 min and stored in aliquots at −80 °C until further use.

Blood cell count, lipid profiles (measured directly by homogenous enzymatic colorimetric method), creatinine, glucose, activity of alanine and aspartate aminotransferases (ALT and AST, respectively) and gamma-glutamyl transferase (GGTP), alpha-fetoprotein (aFP), international normalized ratio (INR), and D-dimer were assayed by routine laboratory techniques. C-reactive protein was determined using immunoturbidimetry (Siemens, Marburg, Germany).

Coagulation factors (F), such as FV, FVII, FVIII, FIX, and FX were measured by 1-stage clotting assays using factor-deficient plasmas (Siemens, Marburg, Germany). Clauss method was used to obtain value of fibrinogen concentration [19]. Antithrombin activity was evaluated using Berichrom Antithrombin III based on thrombin inhibition (Siemens, Marburg, Germany). Free protein S was determined using a latex ligand immunoassay (Instrumentation Laboratory, Milan, Italy). Protein C activity was measured using a chromogenic substrate assay (Siemens, Marburg, Germany).

Tissue plasminogen activator (tPA) was measured by chromogenic assays STA Stachrom α2- antiplasmin and STA Stachrom plasminogen (Diagnostica Stago, Asnieres, France). Plasma plasminogen activator inhibitor-1 (PAI-1) antigen levels were measured by an ELISA (American Diagnostica, Greenwich, CT, USA). Thrombin activatable fibrinolysis inhibitor (TAFI) antigen was determined with an ELISA (Chromogenix, Lexington, MA, USA). Plasma TAFI activity was assessed by a chromogenic assay using the Actichrome Plasma TAFI Activity Kit (American Diagnostica, Greenwich, CT, USA).

### 2.3. Fibrin Clot Properties

Plasma fibrin clot permeability was measured as described [20]. Briefly, 20 mM calcium chloride and 1 U/mL human thrombin (Merck, Kenilworth, NJ, USA) were added to citrated plasma. Tubes containing the clots were connected to a reservoir of a Tris-buffered saline. Its volume flowing through the gels was measured within 60 min. A K_s_, which indicates the average size of pores formed in the fibrin network, with low values indicating tightly packed fibrin structure, was calculated from the equation:(1)$$K_{s}=\frac{Q\times L\times \eta }{t\times A\times \Delta p},\;$$
where Q is the flow rate in time (t), L is the length of a fibrin gel, η is the viscosity of liquid (in poise), t is percolating time, A is the cross-sectional area (in cm^2^) and Δp is the differential pressure (in dyne/cm^2^).

CLT value was measured according to assay described by Pieters et al. [21]. Briefly, citrated plasma was mixed with 20 mM calcium chloride, 0.5 U/mL thrombin (Merck, Kenilworth, NJ, USA), 15 µM phospholipid vesicles (Rossix, Mölndal, Sweden) and 18 ng/mL recombinant tissue plasminogen activator (Actilyse, Boehringer Ingelheim, Ingelheim am Rhein, Germany). The mixture was transferred to a microtiter plate and its turbidity was measured at 405 nm at 37 °C. CLT was defined as the time in minutes from the midpoint from clear to maximum turbidity, to the midpoint in the transition from maximum turbidity to the final baseline turbidity.

All measurements were performed by technicians blinded to the sample origin. The inter- and intra-assay coefficients of variation were <8.0% and 5.6%, respectively.

### 2.4. Statistical Analysis

Categorical variables are presented as numbers and percentages. Continuous variables are expressed as mean ± standard deviation (SD) or median and interquartile range (IQR). Normality was assessed by Shapiro–Wilk test. Equality of variances was assessed using Levene’s test. Differences between groups were compared using the Student’s or Welch’s *t*-test depending on the equality of variances for normally distributed variables. The Mann-Whitney *U*-test was used for non-normally distributed continuous variables. Ordinal variables were compared using the Cochran–Armitage test for trend. Nominal variables were compared by the Pearson’s chi-squared test or Fisher’s exact test if 20% of cells had an expected count less than 5. Depending on assumptions, the Pearson correlation or the Spearman’s rank correlation coefficient was computed to measure the linear association or the monotonic trend between two variables, respectively.

Uni- and multivariate logistic regression analyses were performed to determine independent predictors of low K_s_ (the lowest quartile), and prolonged CLT (the top quartile) in the FO subjects’ group. Due to the numerous variables considered, multivariate models were fitted using forward stepwise regression with the *p* < 0.05 threshold stopping rule. The model for prolonged CLT was adjusted for age and sex while the model for low K_s_ was additionally adjusted for fibrinogen (variables were locked in the models a priori). Results of analyses were expressed as odds ratios (OR) along with 95% confidence intervals (95% CI). Two-sided *p*-values < 0.05 were considered statistically significant. Statistical analyses were performed using JMP^®^, Version 15.2.0 (SAS Institute INC., Cary, NC, USA).

## 3. Results

### 3.1. Demographics of Fontan Patients

The mean age of patients during the FO procedure was 5.2 ± 3.9 years whereas median postoperative time from operation to collection of clinical data and laboratory investigations was 20.5 years [IQR 16.3–22.0]. Most FO patients (60.6%) had preoperative ventricular septal defect in single ventricle physiology and were classified as class II of New York Heart Association’s classification (69.7%). The majority of patients had lateral tunnel performed (92.4%), and in almost half (45.5%), fenestration procedure was performed. Nonsteroidal anti-inflammatory drugs were used by 60.6% of the studied group. In 42.4% of FO patients, hepatics complications occurred just before thromboembolic events (Table 1).

### 3.2. Fontan Patient’s vs. Healthy Control Subjects

Table 2 shows characteristics of FO patients and healthy control subjects at the time of enrollment. FO patients had increased red blood cells (RBC) count, but reduced platelet count (*p* < 0.001 for both comparisons). Biomarkers of liver function, i.e., ALT, AST, and GGTP, were more active in FO group than in controls (*p* < 0.001 for all comparisons). aFP and albumin were also higher in this group of patients (*p* = 0.029 and *p* = 0.009, respectively). Moreover, INR and AST to Platelet Ratio Index were elevated in FO patients, whereas prothrombin time was longer among these patients (*p* < 0.001 for all comparison). Total cholesterol (TC) and low-density lipoprotein (LDL) cholesterol were also decreased in FO group (*p* < 0.001 and *p* = 0.017, respectively). Furthermore, among the FO population D-dimer and endothelin were higher than in the control group (*p* = 0.005 and *p* < 0.001, respectively).

Values are displayed as mean ± standard deviation, number (percentage) or median [interquartile range]. Bold values denote statistical significance at the *p* < 0.050 level. aFP: alpha fetoprotein; ALT: alanine aminotransferase; APRI: aspartate aminotransferase to platelet ratio index; AST: aspartate aminotransferase; BMI: body mass index; CLT: clot lysis time; CRP: C-reactive protein; F: factor; GGTP: gamma-glutamyltransferase; INR: international normalize ratio; K_S_: permeation coefficient; LDL: low-density lipoprotein; PAI-1: plasminogen activator inhibitor-1; PLT: platelets; RBC: red blood cells; PT: prothrombotic time; TAFI: thrombin activatable fibrinolysis inhibitor; TC: total cholesterol; tPA: tissue plasminogen activator; vWF: von Willebrand Factor; WBC: white blood cells.

There was 11.0% reduction of clot permeability, whereas CLT was 8.1% longer in FO patients (*p* = 0.005 and *p* < 0.001, respectively). As far as coagulation factors, FO subjects had decreased FVII and FX and increased FVIII (*p* < 0.001, *p* < 0.001 and *p* = 0.037 respectively). Moreover, FO group also had slightly lower levels of fibrinogen, antithrombin, protein S as well as protein C compared to that of controls (*p* = 0.003, *p* = 0.003, *p* = 0.049 and *p* < 0.001, respectively), however, all variables were within the normal range. The other differences between the compared groups include fibrinolysis protein such as reduced TAFI antigen and activity, whereas tPA level was elevated in the FO group (*p* = 0.042, *p* < 0.001 and *p* < 0.001, respectively). vWF antigen and its activity were also higher in FO patients (*p* < 0.001 for both comparisons).

### 3.3. Fibrin Clot Properties Markers in Fontan Patients

Median K_s_ was 6.5 × 10^−9^ cm^2^ [maximum 9.7 × 10^−9^ cm^2^ and minimum 0.8 × 10^−9^ cm^2^], (Table 2). Low clot permeability was defined as the value in the first quartile, i.e., K_s_ ≤ 5.2 × 10^−9^ cm^2^ (*n* = 16, 24.2%). Pearson’s coefficient test revealed that K_s_ correlated negatively with CLT (r = −0.28, *p* = 0.026) and the risk of K_s_ reduction was increased by longer CLT. Median CLT was 114 min [maximum 197.0 and minimum 66.0], (Table 2). Prolonged CLT was defined as time > 141.0 min, i.e., value in the highest quartile (*n* = 13, 19.7%). CLT correlated with the time between FO operation and investigation (r = 0.29, *p* = 0.019).

### 3.4. Fibrin Clot Properties and Liver-Derived Hemostatic Factors in Fontan Patients

Pearson’s coefficient test revealed that K_s_ correlated negatively with FVIII together with fibrinogen, whereas Spearman’s rank coefficient test showed negative correlations of K_s_ and FIX (r = −0.30, *p* = 0.014, r = −0.41, *p* < 0.001 and r = −0.38, *p* = 0.002, respectively). Univariate logistic regression analysis showed that increased levels of fibrinogen, protein S as well as C, TAFI antigen and its activity, were associated with higher risk of reduced clot permeability (Table 3). The multivariate logistic regression analysis showed, fibrinogen as well as TAFI antigen were associated with reduced K_s_ (Table 3). After adjustment for age, sex, body mass index (BMI) and fibrinogen, TAFI antigen was an independent predictor of higher risk of K_s_ ≤ 5.2 × 10^−9^ cm^2^ (OR 1.041 per 1% increase, 95% CI 1.009–1.081, *p* = 0.011).

Bold values denote statistical significance at the *p* < 0.050 level. ALT: alanine aminotransferase; AST: aspartate aminotransferase; BMI: body mass index; CI: confidence interval; CLT: clot lysis time; CRP: C-reactive protein; F: factor; GGTP: gamma-glutamyltransferase; K_S_: permeation coefficient; LDL: low-density lipoprotein; PAI-1: plasminogen activator inhibitor-1; PLT: platelets; RBC: red blood cells; TAFI: thrombin activatable fibrinolysis inhibitor; tPA: tissue plasminogen activator; vWF: von Willebrand Factor; WBC: white blood cells.

CLT correlated with FIX (r = 0.25, *p* = 0.043). In the univariate regression logistic analysis, higher protein C activity and TAFI antigen levels and activity were related to increased risk of prolonged CLT (Table 3). Multivariate logistic analysis regression showed that protein C was associated with prolonged CLT (Table 3), and after adjustment for age and sex this protein was a predictor of prolonged CLT (OR 1.078 per 1% increase, 95% CI 1.027–1.153, *p* = 0.001).

### 3.5. Fibrin Clot Properties and Liver Function Tests in Fontan Patients

Pearson’s coefficient showed negative correlations between K_s_ and ALT, AST as well as GGTP (r = −0.27, *p* = 0.030, r = −0.25, *p* = 0.048 and r = −0.26, *p* = 0.037, respectively). FO patients which had elevated GGTP activity (*n* = 28, 42.4%) had decreased K_s_, compared to the others with GGTP within the normal range (5.9 × 10^−9^ cm^2^ vs. 6.8 × 10^−9^ cm^2^, *p* = 0.042). Among this group, besides GGTP, 48.1% of patients had abnormally increased levels of ALT, AST, total bilirubin or aFP. Multivariate regression logistic analysis adjusted for age, sex and fibrinogen, revealed that GGTP was a predictor of reduced K_s_ (OR 1.025 per 1% increase, 95% CI 1.005–1.053, *p* = 0.033). Moreover, there was a positive correlation between GGTP and the time between FO surgery and investigation (r = 0.25, *p* = 0.045). In the univariate logistic regression analysis adjusted for age, BMI, sex, and fibrinogen, albumin was associated with decreased risk of K_s_ ≤ 5.2 × 10^−9^ cm^2^ (OR 0.848 per 1 g/L, 95% CI 0.694–0.988, *p* = 0.034). The multivariate analysis model indicated that K_s_ ≤ 5.2 × 10^−9^ cm^2^ was independently associated with albumin concentration (Table 3) even after adjustment for age, sex, BMI and fibrinogen (OR 0.640 per 1 g/L, 95% CI 0.433–0.840, *p* = 0.006).

For the univariate regression logistic analysis, increased levels of LDL cholesterol were associated with increased risk of prolonged CLT (Table 3). Multivariate logistic analysis regression revealed the association between LDL cholesterol and CLT (Table 3). Even after adjustment for age and sex, LDL cholesterol was an independent predictor of prolonged CLT (OR 6.360 per 1 μmol/L increase, 95% CI 1.492–39.894, *p* = 0.011).

### 3.6. Fibrin Clot Properties and Endothelium-Derived Hemostatic Factors in Fontan Patients

Spearman’s coefficient test showed correlation of vWF antigen with K_s_ (r = −0.30, *p* = 0.016). Whereas elevated tPA was associated with lower risk of prolonged CLT even after adjusting for age, sex, BMI, and fibrinogen (OR 0.550 per 1 ng/mL increase, 95% CI 0.314–0.854, *p* = 0.004), (Table 3).

## 4. Discussion

To our knowledge, this study is the first to show associations of hepatic and endothelial abnormalities, observed in adult FO patients, with fibrin clot properties, reflected by clot density, (assessed by K_s_) and impaired fibrinolysis (measured by CLT). Our study revealed that cellular liver damage, reflected by noninvasive liver function tests and endothelial injury were associated with prothrombotic clot phenotype in adults following FO surgery.

Hepatic disorders, which are common in FO patients, are caused by hemodynamic disturbances and systemic venous congestion following FO surgery. This results in a wide spectrum of structural and functional liver alterations. Among many FO survivors these effects progress over time and finally may result in chronic liver disease [10]. In this study we indicated that in FO patients, biomarkers of liver metabolic function—ALT, AST as well as GGTP had elevated circulating activity and together with increased concentrations of total bilirubin and aFP, compared to healthy controls, may suggest hepatic dysfunction. These observations are characteristic for fibrosis without portal hypertension. Moreover, in FO patients we noticed significantly lower platelet count as well as elevated INR that also may indicate progressive liver damage reflected by advanced fibrosis and/or portal hypertension [10,22]. Although the laboratory results of the above-mentioned liver biomarkers were within the reference range, this does not exclude continued progression of detected hepatic changes and their consequences. This suggestion is supported by a positive correlation between GGTP and time following FO procedure and investigation in our study. The dysfunction of the liver, the primary site for synthesis of pro- and anticoagulant proteins, may result in reduced values of most coagulation and fibrinolysis factors, which might manifest by formation of impaired plasma fibrin clot less susceptible to lysis [23].

However, not all factors that play a role in hemostatic process are dependent on liver synthesis. Endothelium is also a source of plasma fibrinolysis proteins such as tPA as well as PAI-1. Moreover, vWF is critical for platelet adhesion in response to endothelial cells activation due to its injury and serves as a protective carrier protein for FVIII [24]. In the FO population, nonpulsative flow in the pulmonary circulation, slow venous flow, and hypoxemia may damage endothelial cells. The simultaneous observation that endothelial injury, reflected by increased plasma level of the mentioned factors is associated with impaired fibrinolysis, could were expected as it corresponds to other studies [18,25,26]. In our study, the value of endothelial-derived tPA was elevated along with significant reduction of liver-derived TAFI, which may predispose to hyperfibrinolysis. Nevertheless, the hemostatic imbalance in FO patients was shifted towards a prothrombotic tendency instead of a predisposition to bleeding. Considering that among the FO group CLT was significantly prolonged, along with reduced K_s_, and thromboembolic complications were one of the most common, right after hepatic difficulties, the probable mechanism of hypofibrinolysis in this study is based on changes in fibrin clot structure.

In this study, we showed abnormalities of most hemostatic factors in the FO population and some of them were related to the analyzed plasma fibrin clot markers. In contrast to other hemostatic factors, the level of FVIII was in the reference range and was slightly higher in FO patients compared to the control group. A significant increase of FVIII was observed in children and adolescents after FO procedures. However, data about level of FVIII in adults are less robust; nevertheless, the recorded levels were within reference range [18,27]. The cause of FVIII elevation in adult patients might be dependent on progressive liver cellular damage [28]. At the same time, the observation of upregulated extrahepatic sites of FVIII synthesis (e.g., kidney, lung, or spleen), in patients with hepatic disorders could also be a probable mechanism behind this phenomenon among the studied FO patients [28]. Further investigation is needed to clearly understand the cause of the detected alteration. Moreover, hypercoagulation, along with forming of more compact clot networks, was observed also in the presence of higher concentrations of FVIII [29,30]. The current study connects these observations, revealing a negative correlation between K_s_ and FVIII and may suggest the possible mechanism of altered fibrinolysis. Simultaneously, we observed changes in levels and activity of fibrinolytic proteins, including the TAFI antigen as well as activity which were slightly lower in FO patients that can at least in part contribute to impair inhibition of fibrinolysis and lead to bleeding. A lack of this fibrinolytic inhibitor is another effect of impaired liver function [31]. In the studied group, bleeding events occurred less frequently than thromboembolic ones, which stands by our finding that alterations in hemostasis have a prothrombotic tendency.

Our novel finding revealed that liver damage among FO adults was related to changes in fibrin clot properties. We showed significant associations between liver dysfunction markers and K_s_ as well as CLT. The most meaningful value had GGTP, the activity of which was strongly associated with K_s_ and was an independent predictor of its reduction. Almost half of the FO group with elevated GGTP activity had significantly reduced clot permeability. It was previously observed that patients with chronic liver disease had as much as three times higher plasma GGTP activity than reference value [32]. This enzyme is characterized by high sensitivity, but low specificity, and its activity increases also in disorders other than hepatic ones, including myocardial infarction or renal failure [33,34].

Unexpectedly, we observed higher concentration of serum albumin in FO adults compared to controls. Only 3% of the studied FO patients had albumin and total protein levels below the reference range, but without nonstandard levels of the other liver biomarkers. This may suggest protein-losing enteropathy, which is common in FO patients and may develop at any time following surgery as a result of chronic venous hypertension, low cardiac output or abnormal lymphatics [35]. Moreover, we observed that albumin level independently predicted lower risk of reduced K_s_ in the FO group. In contrast, the LDL cholesterol was significantly reduced in our studied group and corresponds to a study by Whiteside et al. [36]. The mechanism behind this phenomenon remains unknown although it can be related to liver injury, because synthesis and metabolism of this cholesterol fraction is strongly associated with hepatocytes. We also revealed the predictive role of low LDL cholesterol on prolonged CLT in the FO group. Further research is needed to establish the mechanism underlying modulation of fibrin clot features by these two laboratory variables.

More than half of studied group were taking nonsteroidal anti-inflammatory drugs, which are well-known from their hepatotoxicity in long-term use [37]. Therefore, this could be one of mentioned earlier causes of detected liver cellular damage, reflected by laboratory liver tests. Moreover, widely used aspirin can affect clot permeability. It was previously showed that aspirin alter clot network due to fibrinogen acetylation, which results in render fibrin networks looser and their fibers thicker, leading to lower clot rigidity and enhanced clot lysis [38]. In case of FO patients with reduced K_s,_ the implementation of aspirin as a prevention of thromboembolic complications should be considered.

Our data revealed that altered clot characteristics are associated with hepatic dysfunction and it may be possible to determine the subjects of the FO group in which adverse hepatic events might occur more frequently. The positive correlation between CLT, as well as GGTP, and the time between FO surgery and investigation indicate continuous progression of detected hemostatic and hepatic alterations. Early identification of the subgroup of patients who require constant liver function monitoring, along with assessment of fibrinolytic potential and appropriate anticoagulation prophylaxis, might help reduce the risk of progressive liver cellular damage and occurrence of thromboembolic events.

Our study has several limitations. Firstly, the size of the studied group was limited, although representative of adult FO patients. A higher number of patients would be necessary to draw conclusions regarding the independent predictors of reduced K_s_ and prolonged CLT. We assessed hepatic dysfunction only by noninvasive tests. We did not perform radiological studies of the subjects’ livers, nor liver biopsy, which remains the gold standard for the quantification of liver fibrosis and diagnosis of liver cirrhosis. Portal hypertension was also not assessed. Additionally, the real incidence of thromboembolic events may have been underestimated because we analyzed only symptomatic events. Thrombi are very often asymptomatic in FO patients and the prevalence of thromboembolic events is often underestimated.

In conclusion, we demonstrated for the first time that cellular liver damage, reflected by noninvasive liver function tests, and endothelial injury were associated with prothrombotic clot phenotype. Further and larger studies are needed to validate our findings and clarify mechanisms behind the reported associations.

## Figures and Tables

**Figure 1 jcm-10-05976-f001:**
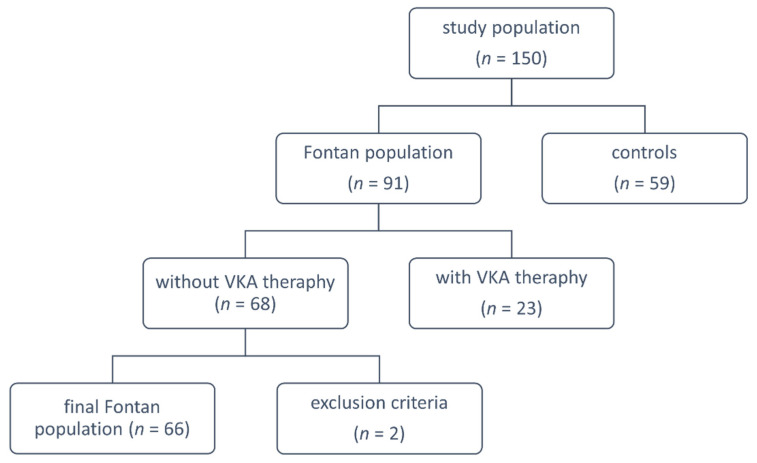
Study flow chart explaining groups for analysis. VKA: vitamin K antagonists.

**Table 1 jcm-10-05976-t001:** Fontan patients’ demographics.

Variable	Fontan Patients (*n* = 66)
Age at Fontan operation (years)	5.2 ± 3.9
Postoperative time (years)	20.5 [16.3–22.0]
Preoperative anatomy, *n* (%)	
Ventricular septal defect in single ventricle physiology	40 (60.6)
Transposition of the great arteries	29 (43.9)
Hypoplasia of right ventricle	19 (28.8)
Tricuspid atresia	15 (22.7)
Double-outlet right ventricle	9 (13.6)
Hypoplastic left heart syndrome	9 (13.6)
Type of Fontan, *n* (%)	
Lateral tunnel	61 (92.4)
Extracardiac conduit	5 (7.6)
Fenestration, *n* (%)	30 (45.5)
NYHA functional class, *n* (%)	
I	16 (24.2)
II	46 (69.7)
III	4 (6.1)
Medication, *n* (%)	
NSAID	40 (60.6)
Beta blockers	11 (16.7)
ACE inhibitors	8 (12.1)
NOAC	10 (15.15)
Complications, *n* (%)	
Hepatic	28 (42.4)
Thromboembolic	8 (12.1)

Values are displayed as mean ± standard deviation, number (percentage) or median [interquartile range]. ACE: angiotensin-converting enzyme; NOAC: novel oral anticoagulants; NSAID: nonsteroidal anti-inflammatory drugs; NYHA: New York Heart Association.

**Table 2 jcm-10-05976-t002:** Basic characteristics of study groups.

Variable	Fontan-Patients (*n* = 66)	Controls (*n* = 59)	*p*-Value
Age in 2018 (years)	23.0 [19.3–27.0]	24.0 [19.0–29.0]	0.565
Male, *n* (%)	43 (65.2)	27 (47.4)	0.845
BMI (kg/m^2^)	21.8 [20.4–24.2]	21.6 [19.2–24.9]	0.821
Resting saturation (%)	93.0 [92.0–94.0]	98.0 [97.0–9.0]	**<0.001**
Laboratory investigations			
RBC (10^6^/mm^3^)	5.3 ± 0.5	4.8 ± 0.5	**<0.001**
WBC (10^3^/mm^3^)	5.7 ± 2.0	6.2 ± 1.5	0.061
PLT (10^3^/mm^3^)	152.0 [120.0–203.8]	237.0 [213.0–266.0]	**<0.001**
TC (μmol/L)	3.7 ± 0.9	4.0 ± 0.6	**<0.001**
LDL cholesterol (μmol/L)	2.3 ± 0.7	2.5 ± 0.6	**0.017**
CRP (mg/L)	1.2 [0.7–2.0]	0.9 [0.7–1.5]	0.211
Creatinine (µmol/L)	72.5 [62.3–82.0]	69 [63.0–85.0)	0.720
ALT (U/L)	23.5 [18.3–31.0]	15.5 [13.0–19.3]	**<0.001**
AST (U/L)	25.0 [21.0–28.8]	19.0 [17.0–22.0]	**<0.001**
Albumin (g/L)	42.5 [40.3–44.6]	40.7 [39.5–43.2]	**0.009**
Total bilirubin (µmol/L)	16.1 [10.9–24.2]	14.0 [12.0–19.0]	0.271
GGTP (U/L)	59.0 [39.0–115.0]	12.0 [7.6–16.1]	**<0.001**
aFP (ng/mL)	2.5 [1.8–4.3]	2.1 [1.6–3.1]	**0.029**
APRI	0.5 ± 0.3	0.2 ± 0.1	**<0.001**
INR	1.4 ± 0.2	1.0 ± 0.1	**<0.001**
PT (s)	15.5 ± 2.1	11.9 ± 0.6	**<0.001**
D-dimer (µg/L)	255.5 [189.8–401.5]	194.0 [170.0–247.0]	**0.005**
Endothelin (pg/mL)	2.5 ± 0.8	1.5 ± 0.3	**<0.001**
Clot characteristics			
K_S_ (x 10^−9^ cm^2^)	6.5 ± 1.8	7.3 ± 1.1	**0.005**
CLT (min)	114.4 [100.5–141.0]	104.8 [87.0–116.4]	**<0.001**
Liver-derived hemostatic factors			
FV (%)	59.0 [40.8–75.3]	79.0 [73.8–82.0]	**<0.001**
FVII (%)	59.5 [52.5–71.0]	96.0 [81.5–105.5]	**<0.001**
FVIII (%)	88.5 [68.0–107.3]	76.0 [64.5–93.0]	**0.037**
FIX (%)	91.5 [78.5–110.5]	87.0 [80.3–104.0]	0.721
FX (%)	85.0 [75.8–92.5]	101.0 [95.0–117.5]	**<0.001**
Fibrinogen (g/dL)	2.4 ± 0.5	2.7 ± 0.5	**0.003**
Antithrombin (%)	94.5 [91.0–101.8]	101.0 [94.5–106.5]	**0.003**
Protein S (%)	90.0 [80.0–100.0]	94.0 [83.5–109.0]	**0.049**
Protein C (%)	97.5 [86.0–112.3]	109 [98.5–125.5]	**<0.001**
TAFI activity (%)	94.3 [81.2–101.6]	101.6 [95.2–109.6]	**<0.001**
TAFI antigen (%)	85.9 [72.3–103.2]	99.2 [83.9–112.2]	**0.042**
Endothelium-derived hemostatic factors			
vWF activity (%)	96.0 [81.8–128.2]	85.7 [62.4–100.6]	**<0.001**
vWF antigen (%)	136.9 [118.6–173.7]	89.9 [71.9–114.4]	**<0.001**
tPA (ng/mL)	5.4 [3.9–6.9]	3.6 [2.5–5.5]	**<0.001**
PAI-1 (ng/mL)	9.9 [6.5–14.7]	9.4 [5.9–13.0]	0.423

Values are displayed as mean ± standard deviation, number (percentage) or median [interquartile range]. Bold values denote statistical significance at the *p* < 0.050 level. aFP: alpha fetoprotein; ALT: alanine aminotransferase; APRI: aspartate aminotransferase to platelet ratio index; AST: aspartate aminotransferase; BMI: body mass index; CLT: clot lysis time; CRP: C-reactive protein; F: factor; GGTP: gamma-glutamyltransferase; INR: international normalize ratio; K_S_: permeation coefficient; LDL: low-density lipoprotein; PAI-1: plasminogen activator inhibitor-1; PLT: platelets; RBC: red blood cells; PT: prothrombotic time; TAFI: thrombin activatable fibrinolysis inhibitor; TC: total cholesterol; tPA: tissue plasminogen activator; vWF: von Willebrand Factor; WBC: white blood cells.

**Table 3 jcm-10-05976-t003:** Associations of plasma fibrin clot properties with clinical variables in Fontan group.

Clinical Variable	Ks ≤ 5.20 × 10^−9^ cm^2^				CLT > 141.0 min			
	Univariate		Multivariate		Univariate		Multivariate	
	Odds Ratio (95% CI)	*p*-Value	Odds Ratio (95% CI)	*p*-Value	Odds Ratio (95% CI)	*p*-Value	Odds Ratio (95% CI)	*p*-Value
Time between surgery and investigation	1.068 (0.941–1.226)	0.315			1.075 (0.938–1.250)	0.302		
RBC	0.566 (0.178–1.647)	0.301			0.847 (0.260–2.629)	0.775		
WBC	1.059 (0.794–1.647)	0.680			0.777 (0.508–1.092)	0.159		
PLT	1.004 (0.995–1.013)	0.366			1.006 (0.997–1.015)	0.213		
LDL cholesterol	1.721 (0.808–3.893)	0.156			3.653 (1.510–11.400)	**0.003**	3.802 (1.114-15.794)	**0.033**
CRP	1.126 (0.978–1.458)	0.104			1.071 (0.933–1.268)	0.288		
ALT	1.011 (0.958–1.063)	0.667			1.004 (0.944–1.058)	0.883		
AST	1.019 (0.945–1.092)	0.605			0.990 (0.899–1.066)	0.801		
Albumin	0.893 (0.752–1.021)	0.105	0.733 (0.547–0.901)	**0.003**	0.892 (0.767–1.026)	0.107		
GGTP	1.004 (0.995–1.015)	0.437			1.008 (0.995–1.020)	0.223		
Endothelin	0.770 (0.313–1.625)	0.511			0.392 (0.109–1.059)	0.067		
K_s_					0.738 (0.511–1.019)	0.066		
CLT	1.026 (1.004–1.052)	**0.021**						
FV	1.006 (0.982–1.029)	0.634			1.004 (0.978–1.029)	0.760		
FVII	1.015 (0.980–1.052)	0.389			1.035 (0.997–1.078)	0.712		
FVIII	1.013 (0.991–1.036)	0.252			0.990 (0.965–1.014)	0.430		
FIX	1.041 (1.014–1.074)	**0.003**			1.015 (0.988–1.042)	0.280		
FX	1.036 (0.997–1.083)	0.074			1.015 (0.974–1.059)	0.465		
Fibrinogen	12.441 (3.238–72.664)	**<0.001**	11.798 (2.658;87.659)	**<0.001**	1.333 (0.403–4.218)	0.627		
Antithrombin	1.026 (0.970–1.089)	0.375			1.035 (0.977–1.102)	0.246		
Protein S	1.058 (1.017–1.111)	**0.004**			1.021 (0.985–1.061)	0.254		
Protein C	1.038 (1.009–1.072)	**0.008**			1.057 (1.023–1.101)	<0.001	1.069 (1.024–1.132)	**0.002**
TAFI activity	1.088 (1.031–1.162)	**0.001**			1.080 (1.021–1.156)	0.005		
TAFI antigen	1.045 (1.016–1.080)	**0.002**	1.040 (1.008;1.078)	**0.012**	1.031 (1.003–1.063)	0.028		
vWF activity	1.014 (0.997–1.032)	0.110			0.996 (0.975–1.014)	0.682		
vWF antigen	1.012 (0.998–1.028)	0.097			0.990 (0.970–1.006)	0.227		
tPA	0.908 (0.691–1.091)	0.359			0.733 (0.502–0.986)	0.038		
PAI-1	1.032 (0.996–1.081)	0.085			0.986 (0.912–1.029)	0.577		

Bold values denote statistical significance at the *p* < 0.050 level. ALT: alanine aminotransferase; AST: aspartate aminotransferase; BMI: body mass index; CI: confidence interval; CLT: clot lysis time; CRP: C-reactive protein; F: factor; GGTP: gamma-glutamyltransferase; K_S_: permeation coefficient; LDL: low-density lipoprotein; PAI-1: plasminogen activator inhibitor-1; PLT: platelets; RBC: red blood cells; TAFI: thrombin activatable fibrinolysis inhibitor; tPA: tissue plasminogen activator; vWF: von Willebrand Factor; WBC: white blood cells.

## Data Availability

The data presented in this study are available on request from the corresponding author.

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
