# Peer review of "Properties of Plasma Clots in Adult Patients Following Fontan Procedure: Relation to Clot Permeability and Lysis Time—Multicenter Study"

_jcm, 2021, doi:10.3390/jcm10245976_

Round 1
Reviewer 1 Report
This manuscript of Skubera et al studied the influence of hepatic and endothelial abnormalities on the risk of thrombotic complications after Fontan procedure. The study design is appropriate. There are few suggestions/concerns that I would like to share.
Comments:
- The authors state that it is a multicenter study. How many centers were involved in the study?
- Table 3 should be moved below the corresponding text
- There are many important data entered in the text that are not found in the tables (for example line 227-228)
- Although the authors acknowledge their limitations, it is difficult for these limitations to be overcome. A higher number of patients would be necessary to draw conclusions regarding the independent predictors of low Ks and prolonged CLT.
- There are several grammatical errors
Author Response
Dear Reviewer of International Journal of Clinical Medicine,
Thank you very much for your patience and all the valuable comments and suggestions. I tried my best to address all your concerns. Please find my answers below.
Sincerely yours,
Dariusz Plicner on behalf of the authors.
- The authors state that it is a multicenter study. How many centers were involved in the study?
Three centers were involved in the study:
- Department of Cardiac and Vascular Diseases, John Paul II Hospital, Krakow
- 1st Department of Cardiology, Poznan University of Medical Sciences, Poznan
- Department of Cardiology, Polish Mother's Memorial Hospital, Research Institute, Lodz
We have added the information to the Materials and Methods section (page 2, line 95).
- Table 3 should be moved below the corresponding text
According to your suggestion we have moved Table 3 below the corresponding text (page 7, line 264).
- There are many important data entered in the text that are not found in the tables (for example line 227-228)
The appropriate changes have been made according to your suggestion (Page 7, Table 3).
- Although the authors acknowledge their limitations, it is difficult for these limitations to be overcome. A higher number of patients would be necessary to draw conclusions regarding the independent predictors of low Ks and prolonged CLT.
Thank you for your valuable comment. We have added the information in Limitations section, according to your suggestion (page 10, lines 429-431).
- There are several grammatical errors.
Appropriate changes have been made, according to your suggestion.
Reviewer 2 Report
Well written paper.
Author Response
Dear Reviewer of International Journal of Clinical Medicine,
Thank you very much for your patience and all the valuable comments and suggestions. I tried my best to address all your concerns. Please find my answers below.
Sincerely yours,
Dariusz Plicner on behalf of the authors.
Well written paper.
Thank you kindly for your comment.
Reviewer 3 Report
Skubera et al. set out to demonstrate the relationships between the properties of plasma clots and changes in clotting factors caused by liver and endothelial cell damage after Fontan (FO) surgery. The authors used Pearson's correlation and Cochran-Armitage test to assess the trends among the above-mentioned subjects. And the author also performed logistic regression analyses to determine the independent predictors of low Ks and prolonged CLT in the FO group.
The findings offer new evidence about cellular liver damage and endothelial injury after FO surgery were associated with prothrombotic clot phenotype. Overall, the manuscript (MS) is a well-written one and the work may be useful to clinical practice to some extent. However, in my personal opinion, there are still the following suggestions to be considered:
Specific comments:
Abstract
- The "Methods" section does not specify the main research methods used in the MS, but most of them are reporting results.
Materials and methods
- A flowchart is needed to illustrate the total number of FO patients and healthy control subjects undergoing health examinations in the hospital, and clearly indicate the number of excluded and remaining patients at each stage of the study.
Results
- All the OR values in Table 4 contain "1", which means that the P-value is not statistically significant. How can the author conclude that the risk is increased or decreased? And the causal relationship between Ks and CLT and liver complications remains to be clarified.
Discussion
- Line 350-355, why does the FO group have high protein levels, but it is explained by postoperative protein-losing enteropathy?
- Line 375,there should be no subjective evaluation in the conclusion, I suggest deleting it.
Minor points:
- The MS should be carefully reinspected since there are some punctuation mistakes (Line 34, Line 38, Line 240, Line 261).
- Line 28-29 did not report the P-value.
- I noticed that the first row of the variables in Table 1 and Table 2 are bolded. If it is for emphasis, the format also needs to be consistent.
- Why is the narrative of the paper non-linear (Table 2, then Table 3, then back to Table 2 again)? This arrangement creates confusion and has to be rearranged.
Author Response
Dear Reviewer of International Journal of Clinical Medicine,
Thank you very much for your patience and all the valuable comments and suggestions. I tried my best to address all your concerns. Please find my answers below.
Sincerely yours,
Dariusz Plicner on behalf of the authors.
Abstract
- The "Methods" section does not specify the main research methods used in the MS, but most of them are reporting results.
The information about research methods was added in the Methods section (page 1, lines 24-26).
Materials and methods
- A flowchart is needed to illustrate the total number of FO patients and healthy control subjects undergoing health examinations in the hospital, and clearly indicate the number of excluded and remaining patients at each stage of the study.
The flow chart has been added to the text, according to your suggestion (page 3).
Results
- All the OR values in Table 4 contain "1", which means that the P-value is not statistically significant. How can the author conclude that the risk is increased or decreased? And the causal relationship between Ks and CLT and liver complications remains to be clarified.
Thank you for your comment. For clarity and better reading purposes, we deleted Table 4.
Discussion
- Line 350-355, why does the FO group have high protein levels, but it is explained by postoperative protein-losing enteropathy?
The majority of the FO population had albumin concentration in the reference range, but 3% had hypoalbuminemia. One of the explanations of hypoalbuminemia is protein-losing enteropathy, which might occur at any time following FO surgery. The appropriate reference has been added, when discussing this phenomenon. The information about postoperative protein-losing enteropathy has been clarified (page 10, lines 400-401).
- Line 375,there should be no subjective evaluation in the conclusion, I suggest deleting it.
We have deleted subjective evaluation in the Conclusion, according to your suggestion.
Minor points:
- The MS should be carefully reinspected since there are some punctuation mistakes (Line 34, Line 38, Line 240, Line 261).
Appropriate changes have been made, according to your suggestion.
- Line 28-29 did not report the P-value.
Appropriate changes have been made, according to your suggestion.
- I noticed that the first row of the variables in Table 1 and Table 2 are bolded. If it is for emphasis, the format also needs to be consistent.
Appropriate changes have been made according to your suggestion.
- Why is the narrative of the paper non-linear (Table 2, then Table 3, then back to Table 2 again)? This arrangement creates confusion and has to be rearranged.
Thank you for your comment. We removed the accidentally placed reference to Table 3 (page 7, lines 264).
- The authors should take into account further factors influencing the results –comorbidities of the liver, drugs taken by the patients and last but not least furthermarkers of primary and secondary haemostasis. Can they provide the informationabout the drugs taken by the patients influencing liver function tests ?
The information about drugs taking by Fontan patients is provide in Table 1. We also have added information about potentially influence of non-steroidal anti-inflammatory drugs, which were mainly used in study population, on the liver functionality and results of liver tests, as well as on fibrin clot structure in Discussion section (page 10, lines 411-419).
- What are the recommendations of the authors for the clinical practice ? How can bethe management of the patients undergoing Fontan surgery changed ?
The clinical practice was described in Discussion section (page 10, lines 420-427).